# Visualizing and Inferring Chromosome Segregation in the Pedigree of an Improved Banana Cultivar (Gold Finger) with Genome Ancestry Mosaic Painting

Alberto Cenci [1,*], Guillaume Martin [2,3], Catherine Breton [1,3], Angélique D'Hont [2,3], Nabila Yahiaoui [2,3], Julie Sardos [1,3] and Mathieu Rouard [1,3,*]

1   Bioversity International, Parc Scientifique Agropolis II, 34397 Montpellier, France
2   CIRAD, UMR AGAP Institut, F-34398 Montpellier, France
3   UMR AGAP Institut, Univ Montpellier, CIRAD, INRAE, Institut Agro, F-34398 Montpellier, France
*   Correspondence: a.cenci@cgiar.org (A.C.); m.rouard@cgiar.org (M.R.)

**Abstract:** Banana breeding faces numerous challenges, such as sterility and low seed viability. Enhancing our understanding of banana genetics, notably through next-generation sequencing, can help mitigate these challenges. The genotyping datasets currently available from genebanks were used to decipher cultivated bananas' genetic makeup of natural cultivars using genome ancestry mosaic painting. This article presents the application of this method to breeding materials by analyzing the chromosome segregation at the origin of 'Gold Finger' (FHIA-01), a successful improved tetraploid variety that was developed in the 1980s. First, the method enabled us to clarify the variety's intricate genetic composition from ancestral wild species. Second, it enabled us to infer the parental gametes responsible for the formation of this hybrid. It thus revealed 16 recombinations in the haploid male gamete and 10 in the unreduced triploid female gamete. Finally, we could deduce the meiotic mechanism lying behind the transmission of unreduced gametes (i.e., FDR). While we show that the method is a powerful tool for the visualization and inference of gametic contribution in hybrids, we also discuss its advantages and limitations to advance our comprehension of banana genetics in a breeding context.

**Keywords:** *Musa*; genome ancestry mosaic painting; meiosis; gametes; breeding; bioinformatics

## 1. Introduction

Bananas are one of the most popular fruits and an important food crop for smallholders in sub-Saharan Africa, Latin America, and Asia [1]. The clonal nature of this crop makes its cultivation very susceptible to pests and diseases [2,3], and banana production has been threatened by multiple pests and diseases, including the devastating Fusarium wilt [4,5].

While most banana cultivars consumed today are natural hybrids, the development of new disease-resistant hybrid varieties has become a primary goal [6–8]. However, edible bananas have inherent genetic features that need to be overcome for efficient breeding strategies. The main obstacles are related to sterility, low seed viability, high heterozygosity, irregular meiotic behavior linked to polyploidy, and large structural chromosome rearrangements [9–11]. Most of the large-scale cultivated edible bananas are triploids (2n = 3x = 33), although there are also diploid varieties, all resulting from hybridizations both within and between wild species [12,13]. Their taxonomic classification has traditionally relied on morphological characteristics that differentiate two main wild species contributing to the cultivated varieties, namely *Musa acuminata* Colla (A genome) and *Musa balbisiana* Colla (B genome) [14]. This classification has arguably led to the identification of genome groups such as AA, AB, BB, AAA, AAB, ABB, and AAAB [14].

Breeding programs have the ultimate objective of developing disease-resistant cultivars that meet the preferences of consumers [6]. To generate bi-parental populations,

breeding schemes involving 2x, 3x, and 4x individuals as male and female parents have been used. Whereas triploids are characterized by low male and female fertility, wild diploids are generally both male- and female-fertile [15]. Two main strategies have been used to date. The first one, denoted here as 3x/2x, has consisted of crossing a female triploid with residual fertility with a diploid male parent, usually an improved diploid bearing some disease resistance. This type of cross generates progenies with various ploidy levels, including some 4x and 3x [6]. It may be followed by a second step (4x/2x cross) where secondary triploids are created through crossing the previously obtained 4x hybrids with wild or improved diploids. A second strategy consists of crossing diploids with tetraploids obtained through chromosome doubling [6]. These strategies may exploit improved diploid parents to introduce desirable new traits obtained from wild relatives, which, thus, extends the timeframe for releasing new cultivars [16]. The conventional breeding cycle in bananas is a long process, with an estimated average time of 10–12 years [17], and few breeding products have been successfully introduced to markets and adopted [8,18].

Some popular hybrids were produced by the Honduran Agricultural Research Foundation (FHIA) in the 1980s. Its breeding program concentrated on improving male parents to give them both disease resistance and good agronomic characters [19], then crossed with triploid cultivars to generate synthetic hybrids, mostly tetraploids, like FHIA-01. The improved cultivar FHIA-01 offered resistance to both Fusarium wilt race 1 and subtropical race 4, as well as resistance to black and yellow Sigatoka (*Pseudocercospora fijiensis* and *Pseudocercospora musae*, respectively). In 1995, the Australian industry introduced this variety as "Goldfinger", to represent an alternative to two Australian industry standards, 'Williams' (AAA, Cavendish subgroup) and 'Lady Finger' (AAB, Pome subgroup), that were affected by Fusarium wilt (Race 1). This variety was marketed as a dessert banana with an apple flavor and proved to be highly productive, exhibiting growth, yield, and bunch characteristics that were equal to, or better than, the standard Cavendish cultivar, 'Williams' [20]. FHIA-01 then was widely distributed to more than 50 tropical and sub-tropical countries.

As previously illustrated, conventional breeding in bananas relies on interploidy crosses; this approach is prone to generate progenies with different ploidy levels and including aneuploids [10,21,22]. The use of molecular cytogenetics techniques (e.g., flow cytometry, GISH) has provided important insight into aneuploidy [23], chromosome pairing during meiotic divisions [24,25], and the genome structure of interspecific cultivars [26–28]. However, these methods are labor-intensive and provide relatively low resolution. In the last decades, like in many other crops, progress in sequencing technologies has boosted the production of genomic resources in bananas, leading to the release of reference genome sequences [29–34]. It has increasingly benefited genebank material characterization [35] and breeding programs [7,36], in particular with the use of SNP markers. It has enabled the development of proof of concept studies for GWAS and genomic selection approaches in bananas [37–39], as well as the detection of large structural variation [10,11,40].

Recently, Martin et al. [41,42] showed that genome ancestry mosaic painting is particularly suitable for inferring the history of the natural, edible banana hybrids that arose some thousand years ago. Such analysis is made possible by the relatively high sequence divergence among wild species (*M. acuminata*, *M. schizocarpa*, and *M. balbisiana*), as well as among subspecies within *M. acuminata*. Furthermore, cultivated bananas resulted from a relatively limited number of sexual events (compared to fully fertile annual crops) that finally led to vegetatively propagated poorly fertile or sterile cultivars. The use of genome ancestry mosaic painting has proven to be a valuable tool in studying the origins of cultivars and banana domestication [41,42] and has helped in studying the pedigree relationships of a few cultivated bananas [43]. As a result, some tools were further implemented to visualize and share this information [44,45]. Until now, this approach has not been applied to breeding material but it may hold significant potential in advancing our understanding of banana genetics and may aid in the breeding of this complex crop. Here, we use a bioinformatic method, referred to in this article as genome ancestry mosaic painting, to further understand banana genetics by analyzing the pedigree of FHIA-01. We then

discuss the advantages and disadvantages of using this type of SNP-based method in a breeding context.

## 2. Materials and Methods

### 2.1. Plant Material

We ordered, from the International Musa Transit Center (ITC), leaf samples from FHIA-01, a synthetic hybrid dessert banana bred in 1988, and other available accessions involved in its pedigree (Table 1). FHIA-01 is a tetraploid plant (4x, AAAB) that was developed from a cross between an interspecific triploid banana (AAB, Pome subgroup) and an improved diploid SH-3142 (AA group) used as male parent. SH-3142 was previously obtained from a diploid cultivar (AA, Pisang Jari Buaya) crossed with an improved diploid (material not available).

### 2.2. Restriction-Site-Associated DNA Sequencing

DNA from each accession was extracted following a 2X CTAB protocol, and restriction-site-associated DNA sequencing (RADSeq) [46] was performed using the PstI restriction enzyme, as we previously applied in [40]. The 300–500 short-insert libraries were sequenced with 91 bp paired-end reads using Illumina HiSeq2000 (Illumina, San Diego, CA, USA) by BGI Hong Kong. At BGI, the raw data were modified via the following two steps: (1) reads polluted by adapter sequences were deleted, and (2) reads that contained >50% low-quality bases (quality value $\leq$ 5) or >10% N bases were removed.

### 2.3. Genome Ancestry Mosaics Painting for Pedigree

Using genome ancestry mosaic painting, we characterized the FHIA-01's pedigree (Table 1). We performed SNP calling and used the VCFHunter suite (https://github.com/SouthGreenPlatform/VcfHunter (accessed on 3 December 2023)) as described in [42] to generate chromosome painting for FHIA-01 and the other accessions involved in its pedigree.

The results of this automated workflow were, when necessary, manually curated to define the ancestry mosaics of unresolved chromosome segments and to infer potential haplotypes. Visualizations were drawn using GeMo [44] and Circos [47].

**Table 1.** List of banana accessions used in the study. Plant material was provided by the International Musa Transit Center (ITC) [48], and more information from passport data is available on the Musa Germplasm Information System (MGIS) [49].

| Accession Code | Accession Name | DOI | Genome Group | Type | Collection |
|:---:|:---:|:---:|:---:|:---:|:---:|
| ITC0504 | FHIA-01 | doi.org/10.18730/9K2MT | AAAB | Improved material | ITC |
| ITC0425 | SH-3142 | doi.org/10.18730/9JXGA | AA | Improved material | ITC |
| ITC0649 | Foconah | doi.org/10.18730/9KBYW | AAB | Cultivar (Pome) | ITC |
| ITC0315 | Pisang Tunjuk | doi.org/10.18730/9JQYH | AA | Cultivar (Pisang Jari Buaya) | ITC |

## 3. Results

### 3.1. Genetic Make-Up of the Goldfinger (FHIA-01) Hybrid and Its Relatives

Visual analyses of the painting showed that FHIA-01 is globally composed of 11 chromosomes from subgenome B inherited from the *M. balbisiana* species, and of 33 chromosomes from subgenome A, from the *M. acuminata* species (Figure 1). Three A/B interspecific genome recombinations were observed in sub-telomeric regions on chromosome arms, locally changing the A/B genome ratio to 4A:0B (chromosomes 3 and 8) as well as to 2A:2B (chromosome 5) (Figure 1A). The subgenome A can be further characterized according to the contributions of the different *M. acuminata* subspecies. The A subgenome of FHIA-01 was mainly contributed by the *M. acuminata* ssp. *banksii* genetic group, *M. acuminata* ssp. *zebrina*, and *M. acuminata* ssp. *malaccensis*. Moreover, the A genome was introgressed with

another species, *M. schizocarpa*, as well as uncharacterized genepools [41,50], one of them likely being *M. acuminata* ssp. *halabanensis* [42].

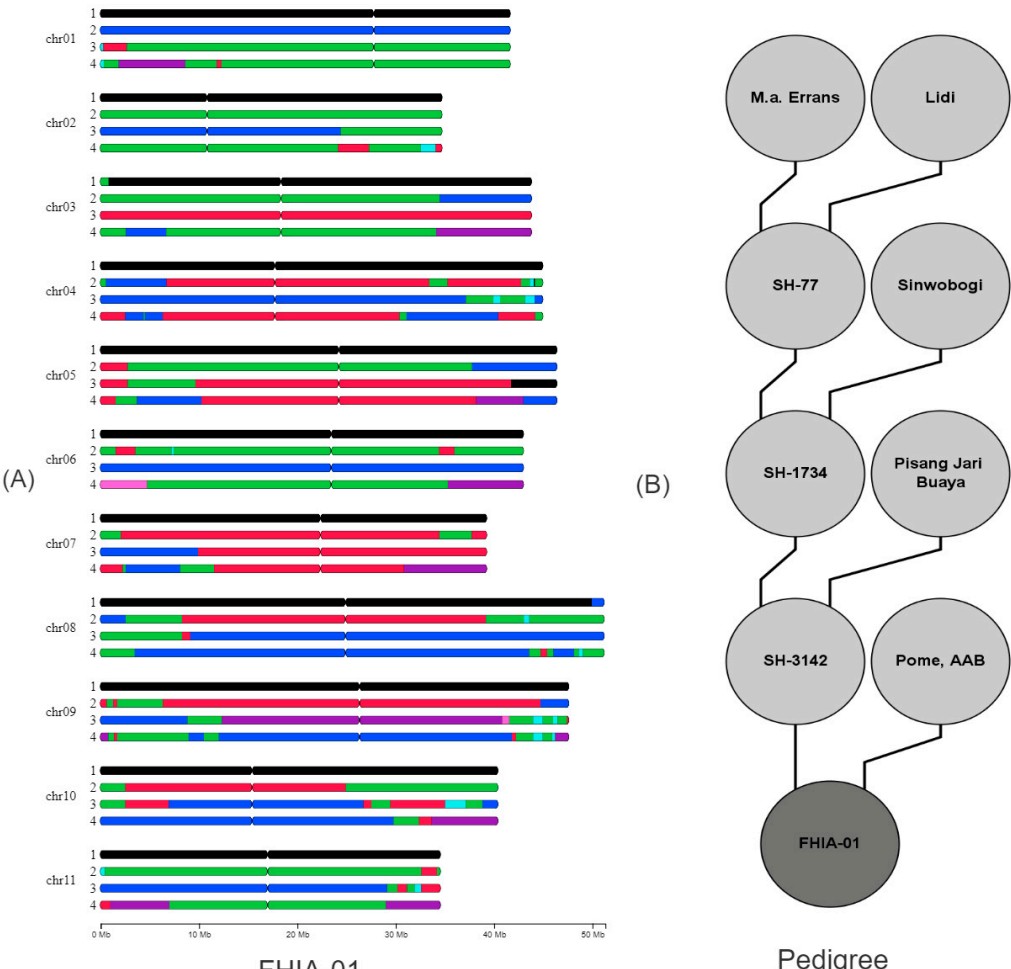

**Figure 1.** (**A**) Genome ancestry mosaic painting applied to the banana accession FHIA-01 and (**B**) full pedigree of FHIA-01. The colors of segments correspond to ancestral contributions (black: *M. balbisiana*, pale-blue: *M. schizocarpa*, green: *M. acuminata banksii* genetic group blue: *M. a. malaccensis*, red: *M. a. zebrina*, pink: uncharacterized genepools, and purple: *M. a. halabanensis*).

FHIA-01′s pedigree is known (Figure 1B) [51]. The male parent of FHIA-01 is the diploid accession 'SH-3142′, for which the genome ancestry painting results are presented in Figure 2. The SH-3142 genome was mainly contributed by *M. acuminata* ssp. *banksii*, *M. acuminata* ssp. *Zebrina*, *M. acuminata* ssp. *malaccensis,* and an uncharacterized genepool inherited from a Pisang Jari Buya cultivar [42], one of the parents of 'SH-3142′. The genome ancestry painting pattern of Pisang Jari Buaya allowed us to infer Pisang Jari Buaya′s gametic contribution to 'SH-3142′ and, therefore, its most probable ancestral mosaic structure (Figure S1).

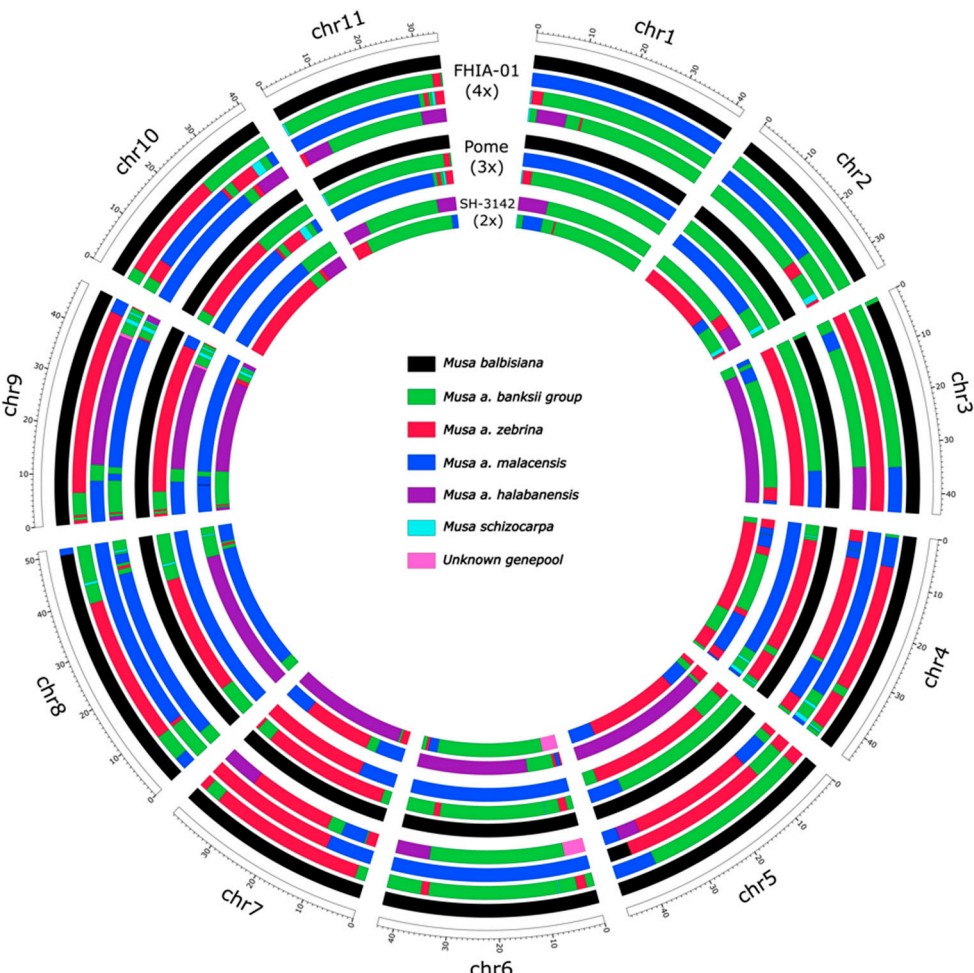

**Figure 2.** Genome ancestry mosaic painting applied to the banana accession FHIA-01 (AAAB, 4x) and its parents Pome (AAB, 3x) and SH-3142 (AA, 2x) in circular mode. The colors of segments correspond to ancestral contributions (black: *M. balbisiana*, pale blue: *M. schizocarpa*, green: *M. acuminata banksii* genetic group, blue: *M. a. malaccensis*, red: *M. a. zebrina*, pink: uncharacterized genepools, and purple: *M. a. halabanensis*).

The female parent of FHIA01 was a 3x = 33 genotype (AAB) within the clonal Pome subgroup. To represent its genotype, the pattern of the 'Foconah' accession was used as a representative of the Pome subgroup, which shares the same mosaic pattern. *M. acuminata* ssp. *banksii*, *zebrina*, and *malaccensis* are the main contributors to the A genomes of Pome, with some introgressions of *M. schizocarpa* and of an uncharacterized genepool on chromosome 9 (Figure 2). The genome ancestry painting of 'Foconah', presented here, confirmed that the Pome genome presents a homoeologous exchange between the B genome and one of the A genomes on the sub-telomeric region of chromosome 3 ([42]—accession 'Prata Ana' of the Supplementary Figures S1 and S2).

### 3.2. Tracing Back Recombination Events in Parental Gametes

By comparing the FHIA-01 mosaic pattern with those of its parents (Figure 2), we were able to infer the ancestral composition of both gametes that contributed to its genetic background. The male gamete from 'SH-3142', with x = 11, was produced via regular meiosis with centromere segregation, and we inferred 16 recombination events along the chromosome arms (as shown in Figure S2).

In contrast, the female gamete 3x = 33 from the AAB genotype, due to its triploid nature, was generated through an irregular meiosis. The results showed that this gamete retained all parental chromosomes except for a few recombination events (Figure 3). This

pattern is consistent with a first-division restitution (FDR), i.e., the absence of chromosome segregation during meiosis I, being at the origin of this gamete. The recombinations observed involved both A/A and A/B chromosome pairs (8 and 2 exchanges, respectively), as illustrated in Figure 3. Notably, and in addition to the A/B exchange on one arm of the chromosome 3B inherited from the Pome parent, two A/B exchanges differentiated the parental genome from the genome of the unreduced gamete. These exchanges involved the sub-telomeric region of A chromosome 5, replaced by its B genome counterpart, and the sub-telomeric region of B chromosome 8, replaced by one of the two A genomes.

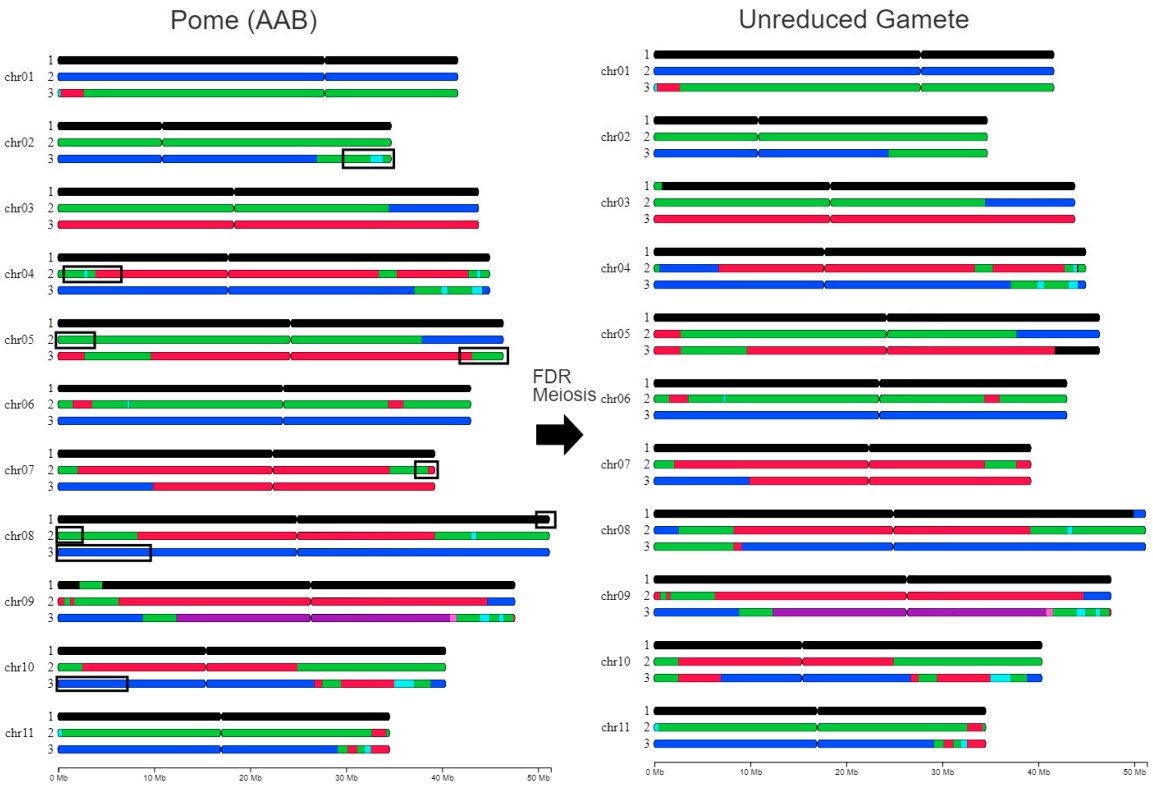

**Figure 3.** Genome ancestry mosaic painting of the triploid AAB genotype of Pome and the triploid female gamete at the origin of FHIA-01. The colors of segments correspond to ancestral contributions (black: *M. balbisiana*, pale blue: *M. schizocarpa*, green: *M. acuminata banksii* genetic group, blue: *M. a. malaccensis*, red: *M. a. zebrina*, pink: uncharacterized genepools, and purple: *M. a. halabanensis*) and black rectangles indicate region where recombinations occurred.

## 4. Discussion

Bananas are a peculiar crop with multiple ploidy levels, for which genetic and genomic studies often require the development of customized scripts or analysis pipelines to accommodate the related specific challenges. Genome ancestry mosaic painting provides an intuitive graphical interface to investigate the genetics of bananas and linked breeding-related considerations. Moreover, it combines an additional layer of information with the inference of the ancestral background of chromosome segments, which may be insightful regarding known and to-be-discovered agronomic specificities.

### 4.1. Visualizing Ancestral Mosaic Patterns in Parents and Progenies

Genome ancestry mosaic painting can be used to visualize ancestral mosaic patterns and unbalanced recombination events involving segments of different ancestries after breeding crosses. Then, the resulting progenies may be selected based on their mosaic painting (Figure 4). Particular genetic patterns resulting from ancestral chromosome segments exchanged during meiosis, anticipated as being favorable, could be identified, selected or tested. For example, some genotypes maximizing the conservation of the

mosaic of one parent, while introgressing a desired trait or QTL from the other parent, could be searched for. Traits of interest for breeders may be associated with regions of specific ancestral origin, as recently shown in the identification of a major QTL-controlling resistance to Subtropical Race 4 [52], and their transmission in crosses can be visualized through genome ancestry painting.

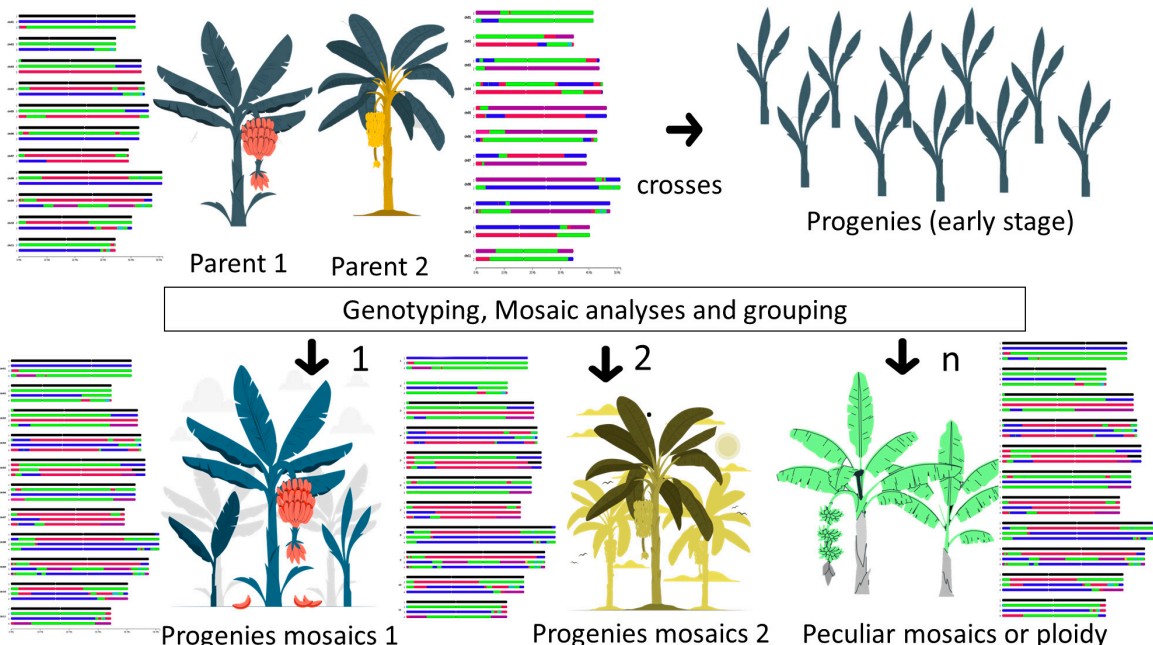

**Figure 4.** Schematic illustration depicting the potential application of genome ancestry mosaic painting in segregating populations within a breeding program. Progeny individuals undergo screening and are grouped based on their recombination events. The breeders then selectively conserve or discard these individuals based on their predefined criteria.

### 4.2. Unreduced Gamete Analysis and Breeding Implications

The specificity of the triploid banana improvement strategies is that they largely build on crosses in which unreduced gametes are produced. The formation of these gametes, exhibiting somatic chromosome numbers, is a common phenomenon in plants that can result from various mechanisms including first-division restitution (FDR), second-division restitution (SDR), or other rare mechanisms [53]. While genome ancestry mosaic painting was originally not designed for this purpose, the graphical visualization of chromosomal mosaics makes it possible to infer the genetic structure of the gametes involved in a progeny. As shown in the tetraploid FHIA-01, the method allowed us to infer both parental gametes by comparing the parental and progeny genomic patterns and to demonstrate the FDR origin of the unreduced parental gamete (Figures 3 and S2). Such an approach enables us to increase our knowledge of banana genetics but not only this, as the identification of the type of unreduced gametes can be important to predicting the efficiency of breeding crosses [54,55]. Indeed, progeny derived from different types of unreduced gamete formation will have different characteristics and, thus, different breeding value depending on the breeding target. Unreduced unrecombined gametes, genetically identical to the parent, may be targeted to preserve the triploid genome of a valued cultivar used as parent or to try reproducing the processes at the origin of successful natural hybrids. For example, the Cavendish banana was shown to bear the complete genome of the Mchare diploid cultivar group [43]. A high level of heterozygosity is generally looked for in bananas [6]. In this context, unreduced gametes with a recombination resulting from FDR, as inferred in the FHIA-01 case, can be of interest. Indeed, such gametes were estimated to conserve 100% parental heterozygosity in the centromeric region and up to 60–70% toward the most distal regions (e.g., [54,56]).

### 4.3. Current Technical Limitations and Perspectives

All types of sequencing techniques generating high-density SNP markers can be employed to implement this method. However, sufficient coverage (approximately 10x by haplotype) is essential for accurate allelic dosage determination. Presently, the method utilizes non-phased SNPs when using short reads, leading to challenges in accurately attributing segments to different haplotypes. Some chromosome segments may not be fully resolved for various reasons, such as marker scarcity or a lack of local intraspecific diversity. In addition, painted segments can sometimes be allocated arbitrarily to a chromosome, as several choices are possible. In such cases, some manual curation might be necessary, and genetic patterns from ancestors in the pedigree have proven to be useful. In the case of FHIA-01, the correct mosaic pattern of the SH-3142 diploid structure was inferred using one of its parents (Figure S1). Looking ahead, SNP calling from long-read technologies will significantly improve haplotype identification resolution, with accurate phasing compared to short reads [57]. Finally, the method does not directly predict recombination events and gamete patterns. Users need to infer such information by observing the transmission of ancestral genome mosaic patterns in successive generations. However, taking advantage of the progress in automatic image recognition and artificial intelligence, it is plausible to automate this step in the future and integrate genome ancestry painting as an additional tool, with other methodologies of pedigree and identity-by-descent analyses as support to banana breeding.

### 5. Conclusions

This study demonstrates the significant potential of genome ancestry mosaic painting in advancing our understanding of banana genetics and in supporting the breeding of this complex crop. The method successfully clarified the intricate genetic composition of the improved tetraploid 'Gold Finger' (FHIA-01). By applying it to different breeding populations, researchers and breeders can gain deeper insights into the genetic diversity, recombination events, and ancestral contributions within these populations. It would thus allow a more comprehensive understanding of the genetic architecture of banana hybrids obtained from crosses of diverse varieties. Genome ancestry mosaic painting could also contribute to deciphering the genetics of desirable traits, such as disease resistance or any other trait of interest. As a prospect, advances in long-read sequencing technology and AI, with its potential for automation and image recognition, hold the key to refining this method further.

**Supplementary Materials:** The following supporting information can be downloaded at: https://www.mdpi.com/article/10.3390/horticulturae9121330/s1, Figure S1. Genome ancestry mosaic painting of the diploid AA genotype of Pisang Jari Buaya and SH-3142; Figure S2. Genome ancestry mosaic painting of the diploid AA genotype of 'SH-3142' and the haploid male gamete at the origin of FHIA-01.

**Author Contributions:** Conceptualization, A.C., M.R. and J.S.; methodology, G.M., N.Y., A.D. and A.C.; formal analysis, A.C., G.M. and C.B.; data curation, A.C.; writing—original draft preparation, M.R. and A.C.; writing—review and editing, J.S., G.M., N.Y. and A.D. All authors have read and agreed to the published version of the manuscript.

**Funding:** This work was partially funded by the Breeding Resources initiative of the CGIAR.

**Data Availability Statement:** The raw sequencing datasets generated and/or analyzed during the current study are publicly available in the Sequence Read Archive (SRA) of the National Center for Biotechnology Information (NCBI) (identifiers: SRR7205563, SRR720568, SRR7205753, SRR7205936).

**Acknowledgments:** We thank the International Transit Center for providing banana samples and BGI for their services for the RAD sequencing. This work was technically supported by the CIRAD—UMR AGAP HPC Data Centre of the South Green Bioinformatics platform.

**Conflicts of Interest:** The authors declare no conflict of interest.

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
