# Peer review of "Visualizing and Inferring Chromosome Segregation in the Pedigree of an Improved Banana Cultivar (Gold Finger) with Genome Ancestry Mosaic Painting"

_horticulturae, doi:10.3390/horticulturae9121330_

Round 1

Reviewer 1 Report

Comments and Suggestions for Authors

The manuscript with the title of “Visualizing and Inferring Chromosome Segregation in the Pedigree of an Improved Banana cultivar (Gold Finger) with Genome Ancestry Mosaic Painting” analyzed the pedigree of FHIA-01, a synthetic hybrid dessert banana, using Genome Ancestry Mosaic Painting method. The genetic make-up of FHIA-01 and its relatives, and the recombination events in parental gametes were analyzed. It will contribute to the understanding of banana genetics. The manuscript is well organized and written. There are a few small comments for the manuscript.

1.     For the Materials and Methods section, the number of “2.2.” (line 98) should be changed to “2.1.”.

2.     For the Materials and Methods section, the titles of section 2.1. and 2.3. are the same. The title of section 2.1. (line 98) is suggested to be revised. The content of the section 2.1. is mainly about the materials used in the study.

3.     For the Materials and Methods section 2.3., the “Figure 1“ (line 114) is suggested to be changed to “Table 1”.

4.     The sizes of figures 1, 2, 3, 4 and S1 are so small that the details (such as words) are not clear. The figures are suggested to be reorganized or enlarged.

5. In the phrase “ genome groups such as AA, AB, BB, AAA, AAB, ABB, AAAB [14].” (line 39), “and” should be added before “AAAB”. 

6. For the Materials and Methods section 2.1., the author should describe the details of banana accession ITC0315, like the introductions of FHIA -01 and SH-3142 in the section. Why do the authors chose the ITC0315, and how about the result about it.

Author Response

Thank you very much for taking the time to review this manuscript. Please find the detailed responses below and the corresponding revisions highlighted in yellow in the re-submitted file.

  1. For the Materials and Methods section, the number of “2.2.” (line 98) should be changed to “2.1.”

ANSWER: Thank you for reporting this mistake. The 2.2. was changed into 2.1 Plant material

  1. For the Materials and Methods section, the titles of section 2.1. and 2.3. are the same. The title of section 2.1. (line 98) is suggested to be revised. The content of the section 2.1. is mainly about the materials used in the study.

ANSWER: Agreed. The 2.2. section was changed into 2.1 Plant material as mentioned above.

  1. For the Materials and Methods section 2.3., the “Figure 1“ (line 114) is suggested to be changed to “Table 1”.

ANSWER: done as proposed

  1. The sizes of figures 1, 2, 3, 4 and S1 are so small that the details (such as words) are not clear. The figures are suggested to be reorganized or enlarged.

ANSWER: Figure 1 was redone with less details and another figure 2, a circos, was added as proposed by reviewer 2. All other figures have been enlarged and should be readable printed in A4 format, while they can be easily zoomed in online.

  1. In the phrase “genome groups such as AA, AB, BB, AAA, AAB, ABB, AAAB [14].” (line 39), “and” should be added before “AAAB”. 

ANSWER: Thank you, this was corrected.

  1. For the Materials and Methods section 2.1., the author should describe the details of banana accession ITC0315, like the introductions of FHIA -01 and SH-3142 in the section. Why do the authors chose the ITC0315, and how about the result about it.

ANSWER: A sentence was added in the Materials and Methods: “SH-3142 was previously obtained from a diploid cultivar (AA, Pisang Jari Buaya) crossed with an improved diploid (material not available).”

In the discussion, we then elaborated on the usefulness of previous parents of the pedigree to improve the “phasing” when several options are possible like done with Pisang Jari Buaya.

Reviewer 2 Report

Comments and Suggestions for Authors

The manuscript presented by the authors provides a comprehensive study on the “Visualizing and Inferring Chromosome Segregation in the Pedigree of an Improved Banana cultivar (Gold Finger) with Genome Ancestry Mosaic Painting”, using a novel methods to visualize the chromosome. However, there are some issues for improvement in the manuscript.

Major comments:

1.      In figure 1, the color is not suitable for visualize the different part of chromosome, change it in more apparently way. Additionally, Advanced Circos plot (https://doi.org/10.1002/imt2.35) is better way to visualize and compare the chromosome.

2.      In figure 2 is no meaningful. Just as on panel in Figure 1A or as the supplementary figure.

3.      In figure 3, the dark chromosome is mean what?

4.      In figure 4, it’s a good graphic abstract. How to defined the selectcriteria?

Author Response

Thank you very much for taking the time to review this manuscript. Please find the detailed responses below and the corresponding revisions highlighted in yellow in the re-submitted file.

Major comments:

  1. In figure 1, the color is not suitable for visualize the different part of chromosome, change it in more apparently way. Additionally, Advanced Circos plot (https://doi.org/10.1002/imt2.35) is better way to visualize and compare the chromosome.

ANSWER: Thank you for the suggestion. Figure 1 was made more readable and Figure 2 is now a Circos as requested, allowing to more easily compare the hybrid with its parents as hey are aligned. however, the size of chromosomes may not appear to scale consistently across different segments. This is due to the concentric layout of the diagram, where chromosomes positioned on the outer, middle, or inner parts of the circle might seem differently sized, even if they are not. This is a reason why we want to provide also figures also in linear mode.

Finally, we conserved the color code as this was done on purpose to remain consistent with previously published banana ancestral mosaics. See https://doi.org/10.1111/tpj.16086 and https://doi.org/10.1093/aob/mcad065

  1. In figure 2 is no meaningful. Just as on panel in Figure 1A or as the supplementary figure.

ANSWER:  Previous Figure 2 is now proposed as Supplementary Figure 2. This figure enables to understand the events along the chromosomes that led to the formation of the gamete of FHIA01, which is not visible from Figure 1. Moreover, contrary to circus, the segment size of mosaics remains the same in linear mode, which can be useful when interpreting the recombination.

  1. In figure 3, the dark chromosome is mean what?

ANSWER:  In Figure 3, the black chromosome corresponds to the Musa balbisiana species. It has been defined in the figure legend.

  1. In figure 4, it’s a good graphic abstract. How to define the select criteria?

ANSWER: The defined criteria are not explicitly detailed on the figure itself as it may vary based on what breeding program priorities. However, some criteria were proposed in the discussion like i) genotypes maximizing the conservation of the mosaic of one parent ii) genotypes with introgression of an ancestral segment known to contain a QTL iii) Rare mosaics of the progenies that may be phenotype in a explorative way.

Reviewer 3 Report

Comments and Suggestions for Authors

See the file 

Comments on the Quality of English Language

see the comments

Author Response

Thank you for taking the time to review the manuscript. Please find a point-by-point response below:

Abstract:

Comment: rebuilt the entire section. Clear and short introduction, clear and short sentences about the methods, short sentences for relevant results, and the perspectives or limits, or both.

ANSWER: the abstract was fully rewritten with more direct style and more concise sentences based on your suggestion.

Introduction

Comment: However, edible bananas have inherent genetic features that need to be overcome for efficient breeding strategies. Is this your opinion?

ANSWER: This is not an individual opinion.  It has been frequently stated that breeding strategies are more challenging in bananas compared to many other crops. In the introduction, we elaborated immediately after this sentence with details supporting the statement and with 3 publications providing evidence.

"The main obstacles are related to sterility, low seeds viability, high heterozygosity, irregular meiotic behavior linked to polyploidy, and large structural chromosome rearrangements [9–11]"

Comment: Lines 40-44: add references to support the ideas.

ANSWER: This idea is well described in a review that we cited in the manuscript. We therefore introduced this reference earlier in the text to support it [15].

Comment: lines 56-69: need more improvements and references.

ANSWER:  The paragraph aims to accurately represent certain aspects of the FHIA breeding program activities from the 1970s, a period characterized by a limited scholarly bibliography. As such, the references we have provided, albeit few, are among the rare peer-reviewed references that document this historical context. From our perspective, the paragraph is presented with clarity, and we believe it effectively conveys useful information within the constraints of available literature.

Materials and methods

Comment: Restriction-site-associated DNA sequencing: reformulate this part and add references.

ANSWER: it was revised with addition of references for the molecular techniques and previous usage of this technology on banana plants.

Discussion

Comment: Need more improvements, particularly in comparison with similar studies in other plants. Equally, you can discuss the importance of the methods employed in your study and your findings.

ANSWER: In contributing to the special issue on banana breeding, our discussion has been more focused to the genetic background of bananas. Then, we acknowledge that the methodology we employ is currently quite distinctive  from other studies for other crops, which presents challenges in finding and comparing. However, we discussed it in a wider context, extending the discussion to citrus, where there are some similarities, and drawing on comprehensive reviews of meiotic behaviors. references 54 -57.

The significance of the method was addressed in earlier publications, references that are included in our citations. Our manuscript, being a brief report type, doesn't go into as much detail as earlier works, but rather aims to extend the conversation into pre-breeding and breeding applications. We hope this will encourage breeders to explore the method's potential as proposed in Figure 4. 

References: improve them

ANSWER: Some references related to comments above were added and we hope this is now satisfactory for the intended audience.

Reviewer 4 Report

Comments and Suggestions for Authors

ABSTRACT

- The abstract should be improved a little by adding some concrete results presented in the paper.

INTRODUCTION

- What are the working hypotheses?

CONCLUSIONS

- The way the paper is structured and the way the research is presented, some conclusions are necessary for the reader to retain the essential elements of the paper.

IN GENERAL

The paper is quite asymmetrical. On the one hand, the introduction stresses the importance of diseases and pests in bananas. On the other hand, in the results and discussion, no reference is made to diseases and pests.

Author Response

Thank you very much for taking the time to review this manuscript. Please find the detailed responses below and the corresponding revisions highlighted in yellow in the re-submitted file.

ABSTRACT

- The abstract should be improved a little by adding some concrete results presented in the paper.

ANSWER: The abstract has been fully revised for enhanced clarity and conciseness. It expands on concrete results of the study.

INTRODUCTION

- What are the working hypotheses?

ANSWER: the working hypotheses are that the genome ancestry mosaics of cultivated bananas reflect their genetic make up and therefore can guide development of breeding hybrids either by trying to conserve mosaics or to detect favorable introgression to be inserted in elite hybrids.

The method was applied to natural cultivars but never in a breeding context before. As suggested we added a sentence to clarify the gap and assumptions.

line 95: Until now, it has not been applied to breeding material as this approach may hold significant potential in advancing our understanding of banana genetics, and potentially aid in the breeding of this complex crop. 

CONCLUSIONS

- The way the paper is structured and the way the research is presented, some conclusions are necessary for the reader to retain the essential elements of the paper.

ANSWER: thank you for the suggestion. We added a conclusion to the manuscript from line 268 to 279.

IN GENERAL

The paper is quite asymmetrical. On the one hand, the introduction stresses the importance of diseases and pests in bananas. On the other hand, in the results and discussion, no reference is made to diseases and pests.

ANSWER: We agree with the comment and would like to add some nuance. The primary focus and background of breeding in this crucial crop has always been on developing pest and disease resistance. Unfortunately, the complex genetics of the crop and limited funding have slowed research progress in identifying QTLs associated with these traits. The potential correlation with introgression from ancestral wild relatives is still in an exploratory phase, but we see promising opportunities with this approach, as mentioned on line 204 with a recent study : "Major QTL-Controlling Resistance to the Subtropical Race 4 [51] and their transmission in crosses can be visualized through genome ancestry painting."

It is only one reference and ideally, we would have like to provide more examples of this kind. but they are missing at the moment. we are unsure that undermining the predominant role that played diseases and pests in bananas in the introduction would mitigate this situation. However, we remain open to further suggestions.

Round 2

Reviewer 2 Report

Comments and Suggestions for Authors

The author's response has been fully addressed my concerns. The quality of the paper has apparently improved. I agree with the publication of this article.

Reviewer 4 Report

Comments and Suggestions for Authors

Congratulations on the experimental work and the development of the paper!